# Systematic Review of Primary Outcome Measurements for Chronic Fatigue Syndrome/Myalgic Encephalomyelitis (CFS/ME) in Randomized Controlled Trials

**DOI:** 10.3390/jcm9113463

**Published:** 2020-10-28

**Authors:** Do-Young Kim, Jin-Seok Lee, Chang-Gue Son

**Affiliations:** 1Department of Korean Medicine, Korean Medical College of Daejeon University, 62, Daehak-ro, Dong-gu, Daejeon 34520, Korea; 95kent@naver.com; 2Institute of Bioscience & Integrative Medicine, Dunsan Oriental Hospital of Daejeon University, 75, Daedeok-daero 176, Seo-gu, Daejeon 35235, Korea; neptune@dju.kr

**Keywords:** chronic fatigue syndrome, myalgic encephalomyelitis, RCT, measurement

## Abstract

Background: Due to its unknown etiology, the objective diagnosis and therapeutics of chronic fatigue syndrome/myalgic encephalomyelitis (CFS/ME) are still challenging. Generally, the patient-reported outcome (PRO) is the major strategy driving treatment response because the patient is the most important judge of whether changes are meaningful. Methods: In order to determine the overall characteristics of the main outcome measurement applied in clinical trials for CFS/ME, we systematically surveyed the literature using two electronic databases, PubMed and the Cochrane Library, throughout June 2020. We analyzed randomized controlled trials (RCTs) for CFS/ME focusing especially on main measurements. Results: Fifty-two RCTs out of a total 540 searched were selected according to eligibility criteria. Thirty-one RCTs (59.6%) used single primary outcome and others adapted ≥2 kinds of measurements. In total, 15 PRO-derived tools were adapted (50 RCTs; 96.2%) along with two behavioral measurements for adolescents (4 RCTs; 7.7%). The 36-item Short Form Health Survey (SF-36; 16 RCTs), Checklist Individual Strength (CIS; 14 RCTs), and Chalder Fatigue Questionnaire (CFQ; 11 RCTs) were most frequently used as the main outcomes. Since the first RCT in 1996, Clinical Global Impression (CGI) and SF-36 have been dominantly used each in the first and following decade (26.1% and 28.6%, respectively), while both CIS and Multidimensional Fatigue Inventory (MFI) have been the preferred instruments (21.4% each) in recent years (2016 to 2020). Conclusions: This review comprehensively provides the choice pattern of the assessment tools for interventions in RCTs for CFS/ME. Our data would be helpful practically in the design of clinical studies for CFS/ME-related therapeutic development.

## 1. Introduction

Chronic fatigue syndrome/myalgic encephalomyelitis (CFS/ME) is a debilitating disease characterized by medically unexplained chronic severe fatigue for at least 6 months along with key symptoms such as unrefreshing sleep, postexertion malaise (PEM), impairments in memory or concentration, and/or orthostatic intolerance [1]. The daily lives of patients are heavily impeded, which leads to unemployment for approximately half of patients and being home- or bed-bound for one quarter [2]. The prevalence of CFS/ME is suggested to be approximately 1–2% worldwide [3], and the annual economic cost for medical care is estimated to be up to USD 10,000 per patient in the US [4].

Although various etiologies of CFS/ME, such as autonomic and neurological dysfunction, abnormalities in mitochondrial function, and aberrant gut microbiota, have been hypothesized, they have not yet been clearly revealed [5]. Recently, this disease has become considered a multisystem neuroimmune disease [1]. To date, various randomized controlled trials (RCTs) for therapeutics have been conducted; however, no effective therapy for CFS/ME exists [6]. Recently, the PACE trial, a large-scale clinical study of cognitive behavior therapy (CBT) and graded-exercise therapy (GET), was reported to be effective for CFS/ME [7]. There is however a fair amount of controversy surrounding this PACE trial, likely due to the debates regarding its efficacy and criticisms by researchers and patients due to judgments of restoration as well as side effects [8].

On the other hand, the absence of objective biomarkers of CFS/ME raises a problem for the actual diagnosis of this illness. In addition, clinical evaluations of treatment responses are also dependent on self-reported assessments of symptom severity, leading to potential trouble during the investigation of new therapeutics [9]. Accordingly, methodologically well-designed tools to assess the valuable responses of treatments for CFS/ME are very important. To date, diverse patient-reported outcome (PRO) measurements have been developed and used to assess fatigue status in clinics, such as the Checklist Individual Strength (CIS) scale, Chalder Fatigue Questionnaire (CFQ), and Multidimensional Fatigue Inventory (MFI) [10,11,12]. Many clinical studies, however, have adopted various fatigue-nonspecific instruments, including the 36-item Short Form Health Survey (SF-36), Clinical Global Impression (CGI), and Sickness Impact Profile-8 (SIP-8) [13,14,15]. In fact, researchers need to carefully review the available measurements and choose the most optimized one for the purpose of their own clinical studies. However, it is not easy for researchers to choose the appropriate measurement instruments for CFS/ME-related studies due to the absence of well-established international guidelines.

To identify the assessment tools that help in the clinical study process for CFS/ME, we comprehensively reviewed the primary measurements used in RCTs and determined changes in the use of these measurements.

## 2. Methods

### 2.1. Data Sources and Search Terms

In accordance with the Preferred Reporting Items for Systematic Reviews and Meta-Analysis (PRISMA) guidelines [16], a systematic literature survey was performed using two electronic literature databases, PubMed and the Cochrane Library, throughout June 2020. The search terms were encephalomyelitis/chronic fatigue syndrome, ME/CFS, encephalomyelitis, ME, chronic fatigue syndrome, CFS, randomized controlled trial, RCT, and clinical trial. The trial type was limited to RCTs, and all languages were included.

### 2.2. Eligibility Criteria

Selected articles for this study were determined by the following inclusion criteria: (1) RCTs or randomized controlled crossover trials, (2) patients with CFS/ME as participants, (3) an evaluation of the efficacy of the intervention for CFS/ME treatment, and (4) fatigue-related measurement or outcome. The exclusion criteria were as follows: (1) articles with no full text and (2) studies without mention of the primary or main outcome. We did not have a criterion based on the number of participants in RCTs.

### 2.3. Data Extraction and Analysis

We extracted data on general features of RCTs, such as the number of participants, age, intervention, and treatment period, along with the primary outcome measurement instrument (subscales, items, range of scores, versions, and application of cutoff scores for recruitment).

As a descriptive analysis, this study did not need to apply statistical analyses. Regarding the treatment period, the mean and standard deviation (SD) are presented.

## 3. Results

### 3.1. General Characteristics of RCTs

A total of 540 articles were initially identified from the PubMed and Cochran databases, and 52 articles met the inclusion criteria for this study (Figure 1). Forty-eight RCTs (92.3%) were performed with adult patients (*n* = 5872), while 4 RCTs (7.7%) were performed with adolescent subjects (*n* = 387). Twenty-six RCTs evaluated the efficacy of pharmacologic interventions, and 27 RCTs were conducted to evaluate nonpharmacologic interventions. The mean treatment period was 15.0 ± 9.3 weeks (Table 1).

In terms of the number of primary outcomes in RCTs, 31 RCTs (59.6%) used a single primary outcome (29 RCTs with adults and 2 RCTs with adolescents). Fifteen RCTs (28.8%) adopted two kinds of main measurements (with adult patients), while six RCTs (11.5%) used three kinds of measurements (four RCTs with adults and two with adolescents) as a primary outcome (Table 1).

### 3.2. Characteristics of Primary Measurements in RCTs

As shown in Figure 2, the 52 RCTs used 17 kinds of methodological instruments, which were classified into survey-based measurements (15 instruments in 50 RCTs) and behavioral measurements (two instruments in four RCTs). All RCTs with adults adopted survey-based measurements, while four RCTs with adolescent patients adopted behavioral (two RCTs) and/or survey-based (two RCTs) measurements (Table 1).

Among the 17 kinds of instruments, the SF-36 was most frequently used (30.8%), followed by the CIS (26.9%), CFQ (21.2%), CGI (13.5%), MFI (11.5%), and SIP-8 (11.5%) (Table 2). Twenty-four RCTs adopted at least one subscale score from these measurement instruments, most commonly the fatigue severity score of CIS (12 RCTs) or physical function score of SF-36 (10 RCTs) (Table 2). Alternatively, these instrument-derived scores were applied as cutoff scores for participant inclusion, such as the fatigue severity score of the CIS (11 RCTs), total score of the SIP-8 (six RCTs), physical function score of the SF-36 (four RCTs), and total score of the CFQ (four RCTs) (Appendix A).

### 3.3. Quinquennial Distribution of Primary Measurements for RCTs

Since the first report of an RCT for CFS/ME using GET in 1997, the number of RCTs has increased, reaching a maximum from 2011 to 2015 (17 RCTs) (Figure 2). In the earliest decade (1996–2005), the CGI was dominantly used as a measurement instrument as the primary outcome in RCTs (6 out of 15 RCTs). During the following decade (2006–2015), the SF-36, CIS, and CFQ were preferred (19 out of 27 RCTs). In the last 5 years (2016–2020), the CIS and MFI have been frequently employed as primary outcome measurements (6 out of 10 RCTs) (Figure 2).

## 4. Discussion

In terms of CFS/ME, a symptom-based approach is a key strategy for not only therapy but also diagnosis because of its unknown etiology [1]. The Centers for Disease Control and Prevention (CDC) recommended symptomatic treatment based on the case definition of the Institute of Medicine (IOM) for providing alternative care for patients [68]. The subjective complaints and comprehension of the PROs are crucial in the diagnostic process as well as in evaluating therapeutic responses in clinical practice for CFS/ME. To provide practical guidance in choosing a suitable measurement in clinical studies for CFS/ME, we analyzed the primary outcome measurements in RCTs conducted to date.

Unlike common guidelines recommending single primary outcome measurement in RCTs [69], 21 (40.4%) of the 52 RCTs employed multiple primary measurements (Table 1). This might be due to the absence of a well-established measurement tool specialized for CFS/ME. Among the 17 tools used in the 52 RCTs, only two behavioral measurements (school attendance rate and the number of steps per day) were adopted in four RCTs that enrolled only adolescent participants (Table 1). It is generally well known that adolescent patients show a poorer school attendance rate than healthy controls [70]. The remaining RCTs (50 RCTs with 15 different tools) employed survey-based PRO measurements, likely for many subjective symptoms or disorders, including migraine, major depressive disorder, or anxiety [71,72,73]. We classified the measurements into two groups: nine nonfatigue specialized tools employed mainly in an earlier decade (1996 to 2005) and eight fatigue-specialized measurements which have been dominant since 2016 (Figure 2).

The SF-36, not specialized for fatigue, is the most frequently used measurement based on our data (16 RCTs) (Table 2). It has been broadly applied for measuring patients’ general health status in reference to health-related quality of life (HRQOL). It is well recognized that the HRQOL of CFS/ME sufferers is notoriously poor and has been linked to a 7-fold higher risk of suicide than healthy controls [74,75]. Therefore, the SF-36, especially the physical functioning subscale, was steadily employed as a primary measurement until 2015, often supportively combined with other fatigue-specialized measurements (10 RCTs), such as the CIS or CFQ. Likewise, the SIP-8 score assessing dysfunction of daily behaviors has been used as part of the primary outcome coupled with fatigue-specialized tools (Appendix A).

In regard to fatigue-specialized instruments, the fatigue severity subscale of the CIS and the total score of the CFQ (11-item version) were dominantly employed (Table 2). Both have been commonly endorsed for the evaluation of psychometric fatigue status in RCTs for CFS/ME and other disorders, including rheumatoid arthritis and fibromyalgia [76]. Both instruments assess not only physical but also mental fatigue status, such as concentration and motivation, and they are known to show a very high correlation in assessing fatigue severity [77]. In particular, the CFQ was employed mostly in trials conducted in the UK (9/11 adoptions), while the CIS was preferred in the Netherlands (12/14 adoptions). On the other hand, the MFI, markedly preferred in recent studies along with the CIS, was originally developed for assessing multifarious fatigue status in patients with cancer [12]. The MFI was one of the measures in the Wichita clinical study assessing over 30 kinds of measurements or parameters for CFS/ME in 2005, and the MFI was proven as a valid measurement [78]. Recently, the MFI was applied in a large-scale study to explore the cytokine signature that showed a positive correlation between serum levels of TGF-β and the severity of CFS/ME [79]. Both the MFI and CIS were created by Dutch researchers and contain 20 nearly identical questionnaire items. However, they have some differences in measurement method strategies: a maximum of 140 points with 7-point scales on the CIS versus a maximum of 100 points with 5-point scales on the MFI (Appendix A). Unlike the CFQ-11, the MFI and CIS adopt both positive and negative questions and measure PEM-related symptoms such as “I am tired very quickly or easily”, which is focused on as one of the recently established hallmark symptoms of CFS/ME [1].

In fact, numerous studies certified the validity and reliability of these commonly used instruments for CFS/ME, such as the CFQ-11, CIS, and MFI [10,78,80], while some researchers have pointed out the ceiling effects of these measurements, especially in clinical trials for treatments [81,82]. They are concerned with the possibility that sufferers of CFS/ME tend to report scores close to maximum, thereby hindering the accurate reflection of treatment response and the baseline condition. Most measurement tools (including CFS/ME-specific instruments) have non-CFS/ME-specific questionnaires, such as “I feel tired” or “I feel weak”, which are frequently complained of among general populations. Accordingly, many trials (most RCTs adopted CIS-based primary outcome) used cutoff scores in the process of participant inclusion (Appendix A). On the other hand, responders to the CFQ-11 will obtain high scores due to comparisons with “usual” or “last well-state”. Because most CFS patients have experienced many years of the disease with fluctuating symptoms, assessment methods involving comparisons to “usual” can hardly reflect not only deterioration in status but also treatment response [2]. Thus, some studies have adopted a modified CFQ-11 as a 10-point Likert scale (from 0 points for healthy conditions to 9 points for the worst status) in RCTs for drug development related to CFS/ME [41].

Although no confirmative pathophysiology of CFS/ME has been identified, some new findings have been highlighted, such as aberrant composition of the gut microbiome and altered serotonergic metabolism within the brain [83,84]. In addition, several studies investigating objective parameters for diagnostic and severity assessments, including elevated levels of TGF-β and nanoelectronic assays, have been conducted [79,85]. One group also found a reduction of red blood cell deformability in patients with CFS/ME [86]. Along with these advances in knowledge, it is necessary that a CFS/ME-specialized measurement instrument be developed to reflect the clinical severity and treatment response and objective biomarkers be discovered to ensure CFS/ME.

## 5. Conclusions

This systematic review provides a comprehensive overview of the choice of primary measurements in RCTs for CFS/ME to date. Approximately 40% of RCTs applied multiple primary measurements. Of the 17 kinds of measurement tools, the SF-36 (nonfatigue specific measurement) had been most frequently applied through 2015, while two fatigue-specific measurements, the CIS and MFI, have been frequently employed in recent trials. Our data will be helpful in the practical design of clinical studies for CFS/ME-related therapeutic development.

## Figures and Tables

**Figure 1 jcm-09-03463-f001:**
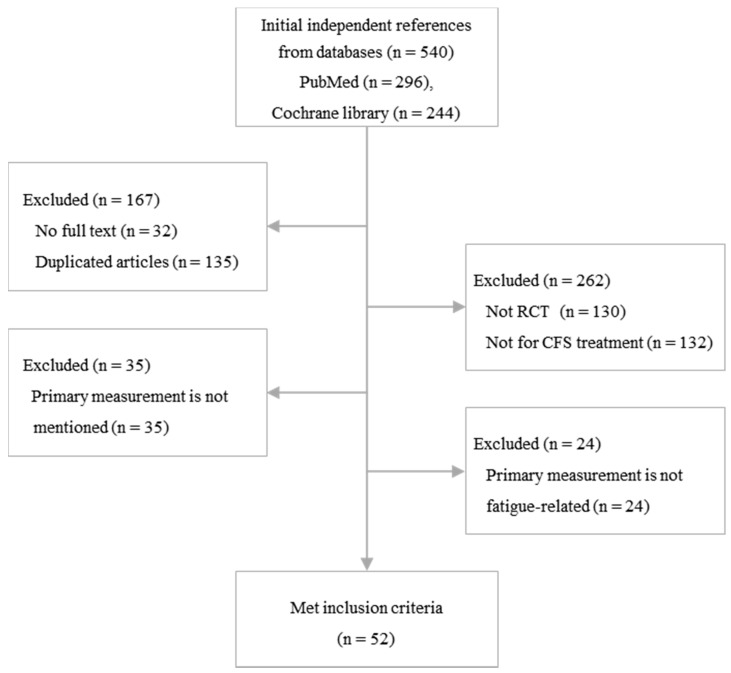
Flow chart of the study.

**Figure 2 jcm-09-03463-f002:**
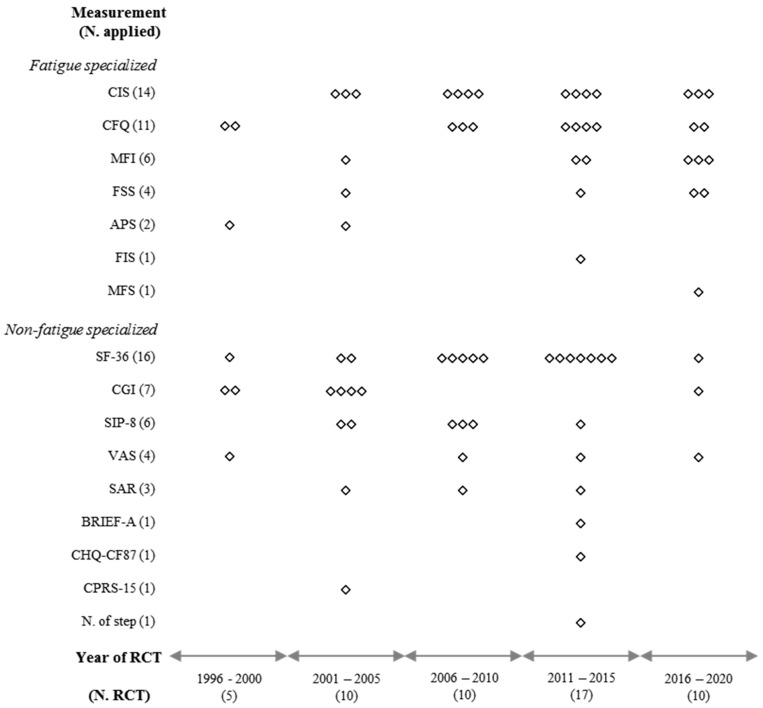
Graphical display for quinquennial distribution of primary measurements.

**Table 1 jcm-09-03463-t001:** Study characteristics.

Items	Adults	Adolescents	Total
^A^ Total RCT (%)	48 (92.3)	4 (7.7)	52 (100.0)
N. of participants (female)	5872 (4437)	387 (302)	6259 (4739)
Treatment period (mean weeks)	14.1 ± 8.2	25.3 ± 16.4	15.0 ± 9.3
RCT for pharmaceutical intervention (%)	25 (96.2)	1 (3.8)	^B^ 26 (100.0)
RCT for nonpharmaceutical intervention (%)	24 (88.9)	3 (11.1)	^B^ 27 (100.0)
N. of primary outcomes in RCT (%)			
One primary measurement	29 (93.5)	2 (6.5)	31 (100.0)
Two primary measurements	15 (100.0)	0 (0.0)	15 (100.0)
Three primary measurements	4 (66.7)	2 (33.3)	6 (100.0)
Methodological classification of measurement instrument (%)			
RCT with behavioral measurement	0 (0.0)	4 (100.0)	4 (100.0)
Kinds of measurement	0 (0.0)	2 (100.0)	2 (100.0)
RCT with survey-based measurement	48 (96.0)	2 (4.0)	50 (100.0)
^C^ Kinds of measurement	14 (93.3)	3 (20.0)	15 (100.0)
Criteria for participant recruitment (%)			
^D^ RCT used measurement-based cutoff score	17 (94.4)	1 (5.6)	18 (100.0)
Behavioral score	0 (0.0)	1 (100.0)	1 (100.0)
^C^ Survey-based score	6 (85.7)	2 (28.6)	7 (100.0)

^A^ The detailed information for the whole RCT list is summarized in Appendix A. ^B^ One RCT used both pharmacologic and nonpharmacologic interventions (fluoxetine + graded exercise therapy). ^C^ Some items were applied multiple times; thus, the total percentage was larger than 100%. ^D^ Eighteen RCTs applied a cutoff score for inclusion criteria.

**Table 2 jcm-09-03463-t002:** Measurement instruments in RCTs.

Scale	N. of RCTs (%)	Measurement and Structure of Items	Subscale for Primary Outcome (N. of RCTs) (Reference)
SF-36	16 (30.8)	Assess functional impairment in eight areas summarized as physical and mental function.	Total score (4) [17,18,19,20] Physical function (10) [7,21,22,23,24,25,26,27,28,29] Physical + social function (1) [30] ^A^ Mental health summary (1) [31]
CIS	14 (26.9)	Covers several aspects of fatigue, such as severity, concentration, motivation, and physical activity.	Total score (2) [32,33] Fatigue severity (12) [17,23,25,28,30,34,35,36,37,38,39,40]
CFQ	11 (21.2)	Measures fatigue severity with items categorized into physical and mental fatigue.	Total score (10) [7,21,24,26,27,41,42,43,44,45] Physical score (1) [46]
CGI	7 (13.5)	Provides a global rating of illness severity and improvement through single Likert scale.	Single item score (7) [44,47,48,49,50,51,52]
MFI	6 (11.5)	Evaluates five dimensions of fatigue: general, physical, mental fatigue, reduced motivation, and reduced activity.	Total score (4) [53,54,55,56] General fatigue (2) [57,58]
SIP-8	6 (11.5)	Measures dysfunction through everyday behavior. Items are summarized into psychosocial, physical, and independent dimensions.	Total score (6) [23,25,37,38,39,40]
FSS	4 (7.7)	Measures the severity of fatigue and its effect on a person’s activities and lifestyle, with no particular subscales.	Total score (4) [53,59,60,61]
VAS	4 (7.7)	Through Likert scale items, assesses subjective characteristics that cannot be directly measured.	Single item score (2) [33,53] Fatigue symptom (2) [20,62]
SAR	3 (5.8)	Measures overall health status of adolescents with school attendance rate.	Attendance rate (3) [28,36,63]
^B^ APS	2 (3.8)	Measures fatigue status through survey designed by authors of RCT according to 1994 CDC criteria.	Fatigue severity (1) [64] Symptom checklist (1) [39]
BRIEF-A	1 (2.0)	Evaluates an adult’s executive functions of self-regulation in everyday environment of patient. Subscales are summarized in behavioral regulation and metacognition.	Total score (1) [65]
CHQ-CF	1 (2.0)	Measures health-related quality of life for adolescents. Items are within 9 subscales that focus on neurologic and psychologic domains.	Physical function (1) [36]
CPRS-15	1 (2.0)	Assesses severity of psychiatric symptoms and observed behavior. Items focus on symptoms of mental disorder.	Total score (1) [50]
FIS	1 (2.0)	Measures the symptom of fatigue as part of an underlying chronic disease or condition. Items are divided into cognitive, physical, and psychosocial functioning.	Total score (1) [66]
MFS	1 (2.0)	Has strength in evaluating psychiatric symptoms with no subcategorization of 15 items.	Total score (1) [47]
N. of steps	1 (2.0)	Evaluates general health status by measuring the number of steps per day.	Number of steps per day (1) [67]

^A^ Mental health summary includes vitality, social function, role function/emotional, and mental health subscales. ^B^ There were 2 kinds of nonestablished measurements which authors of RCTs produced. APS: author-produced scale, BRIEF-A: Behavior Rating Inventory of Executive Function—Adult version, CFQ: Chalder Fatigue Questionnaire, CGI: Clinical Global Impression, CHQ-CF: Child Health Questionnaire—Child Form, CIS: Checklist Individual Strength, CPRS-15: Comprehensive Psychopathological Rating Scale-15, FIS: Fatigue Impact Scale, FSS: Fatigue Severity Scale, MFI: Multidimensional Fatigue Inventory, MFS: Mental Fatigue Scale, SAR: school attendance rate, SF-36: 36-item Short Form Health Survey, SIP-8: Sickness Impact Profile-8, VAS: visual analog scale.

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
