# Peer review of "Systematic Review of Primary Outcome Measurements for Chronic Fatigue Syndrome/Myalgic Encephalomyelitis (CFS/ME) in Randomized Controlled Trials"

_jcm, 2020, doi:10.3390/jcm9113463_

Round 1

Reviewer 1 Report

The review article by DY Kim et al,” Systematic review of primary outcome measurements for chronic fatigue syndrome/myalgic encephalomyelitis (CFS/ME) in randomized controlled trials” comprehensively discusses the choice of assessment tools for interventions in RCTs for CFS/ME. This data may be useful in designing the clinical studies and developing new therapeutic strategy related to CFS/ME.     

The article is straightforward, convincing. However, few important points that could be considered:

-Due to unknown etiology, the diagnosis and therapeutic interventions of ME/CSF is quite puzzling. Since there is no specific laboratory test to diagnose ME/CSF directly and other conditions that can cause similar symptoms, it is essential that authors must have considered in-debt evaluation of symptoms and medical history of patients to ensure ME/CSF during the search of literature for this study.                              

-Typos such as period (.) at the end of line 33 and 65 may be used.

Author Response

The review article by DY Kim et al,” Systematic review of primary outcome measurements for chronic fatigue syndrome/myalgic encephalomyelitis (CFS/ME) in randomized controlled trials” comprehensively discusses the choice of assessment tools for interventions in RCTs for CFS/ME. This data may be useful in designing the clinical studies and developing new therapeutic strategy related to CFS/ME.     

The article is straightforward, convincing. However, few important points that could be considered:

-Due to unknown etiology, the diagnosis and therapeutic interventions of ME/CSF is quite puzzling. Since there is no specific laboratory test to diagnose ME/CSF directly and other conditions that can cause similar symptoms, it is essential that authors must have considered in-debt evaluation of symptoms and medical history of patients to ensure ME/CSF during the search of literature for this study.   

â–¶ We sincerely appreciate to reviewer for important opinion. We deeply agree with necessary of CFS/ME-specialized diagnostic tools or methods, so we added and emphasized need of objective biomarkers to ensure CFS/ME along with disease specialized measurement instruments in discussion (line 199-201). We thank again to reviewer for helpful and professional suggestion.

-Typos such as period (.) at the end of line 33 and 65 may be used.

â–¶ We really thank for the detail review. We corrected the typos by adding period at the end of line 33 and 67(65).

Reviewer 2 Report

The authors reviewed the outcomes of several ME/CFS controlled trials. They aim to show the choice pattern of assessment tools for interventions in randomized controlled trials for ME/CFS. The overall writing quality is good. However, the authors need to address the following issues before publication can be recommended:

  1. There has been a fair amount of controversy surrounding the PACE trials. There is no issue with the authors mentioning the trial, but something as important and as widely commented upon needs to be better discussed in the manuscript.
  2. The authors only talk about Fukuda criteria, but do not talk about studies using the CCC or IOM criteria, which are much modern and restrictive.
  3. A requirement for someone to be diagnosed with ME/CFS is to have the disease for six months (PMID: 11694477). The authors should include this information.
  4. The authors did not mention biophysical assessments of blood cells. For example, RBC deformability was reduced in ME/CFS patients (PMID: 30594919). It is a relevant objective diagnostic study.

Author Response

The authors reviewed the outcomes of several ME/CFS controlled trials. They aim to show the choice pattern of assessment tools for interventions in randomized controlled trials for ME/CFS. The overall writing quality is good. However, the authors need to address the following issues before publication can be recommended:

1. There has been a fair amount of controversy surrounding the PACE trials. There is no issue with the authors mentioning the trial, but something as important and as widely commented upon needs to be better discussed in the manuscript.

â–¶ We sincerely appreciate for the helpful suggestion. We revised the mention about PACE trial and its criticism issues in introduction (line 51-53).

2. The authors only talk about Fukuda criteria, but do not talk about studies using the CCC or IOM criteria, which are much modern and restrictive.

â–¶ We really appreciate for the critical review and professional suggestion. Although the case definition of the disease has been newly revised, and we sensitively recognize, most of included RCTs in our study, even conducted very recently, had used Fukuda criteria for recruitment case definition (Supplementary Table 1). Therefore, according to our main question, we provided an overview of primary measurements in RCTs and there is not any intention of emphasizing certain criteria.

3. A requirement for someone to be diagnosed with ME/CFS is to have the disease for six months (PMID: 11694477). The authors should include this information.

â–¶ We sincerely appreciate for detail review the helpful suggestion. We added the information in the line 38, defining the disease in introduction of manuscript.

4. The authors did not mention biophysical assessments of blood cells. For example, RBC deformability was reduced in ME/CFS patients (PMID: 30594919). It is a relevant objective diagnostic study.

â–¶ We really thank for the helpful suggestion. We recognize importance of newly discovered biophysical features of ME/CFS which could be clue for objective biomarker of the disease. Therefore, we added study of RBC deformability emphasizing investigations of CFS/ME-specialized parameters in discussion (line 198-199).

Round 2

Reviewer 2 Report

The authors have satisfactorily responded to the questions.